# Protection Elicited by Glutamine Auxotroph of *Yersinia pestis*

**DOI:** 10.3390/vaccines13040353

**Published:** 2025-03-26

**Authors:** Svetlana V. Dentovskaya, Anastasia S. Vagaiskaya, Mikhail E. Platonov, Alexandra S. Trunyakova, Ekaterina A. Krasil’nikova, Elizaveta M. Mazurina, Tat’yana V. Gapel’chenkova, Nadezhda A. Lipatnikova, Rima Z. Shaikhutdinova, Sergei A. Ivanov, Tat’yana I. Kombarova, Florent Sebbane, Andrey P. Anisimov

**Affiliations:** 1Laboratory for Plague Microbiology, Especially Dangerous Infections Department, State Research Center for Applied Microbiology and Biotechnology, 142279 Obolensk, Russia; vagaiskaya.anastasiya@gmail.com (A.S.V.); platonov@obolensk.org (M.E.P.); sasha_trunyakova@mail.ru (A.S.T.); ek.al.krasilnikova@gmail.com (E.A.K.); elizavetamazurina99@yandex.ru (E.M.M.); tgapelchenkova@mail.ru (T.V.G.); n.a.lipatnikova@mail.ru (N.A.L.); shaikhutdinova@yandex.ru (R.Z.S.); sa-ivanov@yandex.ru (S.A.I.); anisimov@obolensk.org (A.P.A.); 2Laboratory of Biomodels, State Research Center for Applied Microbiology and Biotechnology, 142279 Obolensk, Russia; kombarova.tatyana@yandex.ru; 3Univ. Lille, CNRS, Inserm, CHU Lille, Institut Pasteur Lille, U1019—UMR 9017—CIIL—Center for Infection and Immunityof Lille, F-59000 Lille, France; florent.sebbane@inserm.fr

**Keywords:** *Yersinia pestis*, bubonic plague, plague vaccine, glutamine synthetase, knockout, guinea pig, mouse

## Abstract

**Background/Objectives:** *Yersinia pestis* is an important zoonotic pathogen responsible for the rare but deadly disease of people with bubonic, septic, or pneumonic forms of plague. The emergence of multidrug-resistant *Y. pestis* strains has attracted more and more researchers’ attention to the search for molecular targets for antivirulence therapy, including anti-nutritional-virulence therapy. The *glnALG* operon plays a crucial role in regulating the nitrogen content within a bacterial cell. This operon codes for three genes: the structural gene *glnA* and the two regulatory genes *glnL* and *glnG*. In this study, we tested the effect of the deletion of glnA and *glnALG* on the pathogenic properties of *Y. pestis*. **Methods:** To assess the contribution of nitrogen metabolism to *Y. pestis* virulence, knockout mutants Δ*glnA* and Δ*glnALG* were constructed. The former was unable to synthesize glutamine, while the latter was not only defective in glutamine synthesis but also lacked the two-component sensor–transcriptional activator pair GlnL and GlnG, which could partially compensate for the decrease in intracellular glutamine concentrations by transporting it from the host or by catabolic reactions. For vaccine studies, immunized mice and guinea pigs were injected s.c. with 200 LD_100_ of the wild-type *Y. pestis* strain. **Results:** A single knockout mutation in the *glnA* gene did not affect the virulence of *Y. pestis* in mice and guinea pigs. Knockout of the entire *glnALG* gene cluster was required for attenuation in these animals. The Δ*glnALG* strain of *Y. pestis* did not cause death in mice (LD_50_ > 10^5^ CFU) and guinea pigs (LD_50_ > 10^7^ CFU) when administered subcutaneously and provided 100% protection of animals when subsequently infected with 200 LD_100_ of the *Y. pestis* virulent wild-type strain 231. **Conclusions:** *Y. pestis*, defective in both the glutamine synthetase GlnA and the two-component sensor–transcriptional activator pair GlnL-GlnG, completely lost virulence and provided potent protective immunity to mice and guinea pigs subsequently challenged with a wild-type *Y. pestis* strain, demonstrating the potential use of the *glnALG* operon as a new molecular target for developing a safe and efficient live plague vaccine.

## 1. Introduction

The continued existence and multiplication of bacteria depend on their ability to adjust to variations in the availability of major, minor, and trace elemental nutrients. Nitrogen is especially important because it is a key ingredient in the building of the majority of biomolecules essential for bacterial cells [1]. The central molecules in nitrogen metabolism are glutamine and glutamate. Glutamine serves as a nitrogen donor for many nitrogen-containing molecules in the cell and is a part of several metabolic processes that promote cell growth and proliferation. Glutamine is considered the main intracellular signal for nitrogen availability in different bacteria [2].

The key enzyme in the nitrogen metabolic pathway is glutamine synthetase (GlnA), which utilizes the substrate L-glutamate and adenosine triphosphate (ATP) to produce glutamine [1]. Under conditions of low external nitrogen availability, glutamine synthetase is the enzyme responsible for ammonia assimilation in its entirety. Glutamine synthetase’s synthesis and activity are controlled in accordance with the levels of available nitrogen. The adaptive response to changes in the extracellular nitrogen content in bacteria is coordinated by a two-component GlnLG system (also called NtrBC) [3]. GlnL (NtrB) is a sensory histidine kinase that perceives and then transduces a nitrogen starvation signal, during which its phosphorylation and subsequent activation of the DNA-binding transcription factor GlnG (NtrC) occurs. The *glnA*, *glnL*, and *glnG* genes form an operon, which is central to nitrogen metabolism. In addition to synthesizing glutamine, enterobacteria can meet their needs for this nutrient by absorbing it from the environment with the help of the ABC glutamine transporter encoded by the gene cluster *glnHPQ* that is also regulated by the two-component GlnLG system together with the alternate sigma factor δ^54^ [4].

GlnA is essential for antimicrobial resistance in many pathogens [5,6,7,8,9]. In *Mycobacterium tuberculosis*, *glnA* was predicted in silico as a possible target for antibacterial substances [10]. In addition, GlnA exhibits moonlighting functionality in *Bacillus subtilis* [11]. The *glnALG* operon deletion [12] or even gene *glnA-1* knockout [13] caused significant attenuation of several other bacterial pathogens. P. Aurass et al. [14] demonstrated that the *glnA* gene is crucial for the growth of *Salmonella* Typhimurium. Furthermore, it regulates significant factors contributing to bacterial virulence, which is contingent upon the presence of glutamine.

*Y. pestis* is an important zoonotic pathogen responsible for the rare but deadly disease of people with bubonic, septic, or pneumonic forms of plague. The emergence of multidrug-resistant *Y. pestis* strains has attracted the attention of more and more researchers to the search for molecular targets for vaccination [15,16], including anti-nutritional-virulence therapy [17]. Our understanding of the specific metabolic and nutritional processes utilized by *Y. pestis* during infection remains limited. Developing more vaccine candidates increases the chances of generating a safe, highly protective, and long-acting vaccine. Recently, it was shown that PhoP-PhoQ and OmpR-EnvZ systems were the only 2 of 23 *Y. pestis’* two-component regulatory systems (2CSs) required for the development of lethal plague infection. Knockout of *glnLG* genes did not reduce the virulence of *Y. pestis* mutants [18]. Our interest in *Y. pestis* glutamine synthetase GlnA arose after several passages of an initially avirulent *Y. pestis* strain I-3189 (subsp. *microti* bv. Ulegeica) for guinea pigs resulted in a 4-fold increase in the level of *glnA* gene expression, which was accompanied by a 1.5 × 10^4^-fold increase in virulence [19] and the appearance of a GlnA protein spot on two-dimensional protein electropherograms [20].

In this study, we generated Δ*glnA* and Δ*glnALG Y. pestis* isogenic mutants to examine the degree of their attenuation and to conduct a comparative assessment of their safety and protective potency. This research demonstrates that deleting both the glutamine synthetase GlnA and the two-component sensor-transcriptional activator pair GlnL-GlnG from *Y. pestis* dramatically reduces the mutant’s virulence. Furthermore, animals immunized with this attenuated strain were strongly protected against death after subsequent infection with a wild-type *Y. pestis* strain. These findings suggest that the *glnALG* operon could be a promising target for the development of a safe and effective live plague vaccine.

## 2. Materials and Methods

### 2.1. Bacterial Strains, Plasmids, and Culture Conditions

The wild-type virulent *Y. pestis* 231 and its derivative mutant strains listed in Table 1 were obtained from the State Collection of Pathogenic Microbes and Cell Cultures at the State Research Center for Applied Microbiology and Biotechnology (SCPM-O). *Escherichia coli* cultures were maintained in a Lysogeny Broth (LB) medium, consisting of 1% peptone, 0.5% yeast extract, and 0.5% NaCl, and incubated at a temperature of 37 °C, and *Y. pestis* strains were grown in BHI (Brain Heart Infusion, HiMedia, Maharashtra, India) at pH 7.2. When needed, the media were supplemented with ampicillin (100 μg/mL, Ap), chloramphenicol (20 μg/mL, Cm), polymyxin B (50 μg/mL, Pol), or L-Glutamine (20 mM; product no. BCBC6452V; Sigma, Saint-Louis, MO, USA).

### 2.2. Animals

The study utilized a cohort of 132 six-week-old outbred mice obtained from the SCRAMB breeding unit in the Moscow Region, Russia. Additionally, 126 four-week-old guinea pigs were procured from the Lab Animals Breeding Center (Russian Academy of Medical Sciences, Stolbovaya, Moscow Region, Russia).

This research adhered to all ethical guidelines regarding animal welfare. It was conducted in accordance with the National Institutes of Health Animal Welfare Assurance #A5476-01, granted on 7 February 2007, as well as the regulations and directives set forth by the European Union for the responsible handling, care, and protection of laboratory animals (https://environment.ec.europa.eu/topics/chemicals/animals-science_en, accessed on 29 January 2024).

### 2.3. Mutagenesis

The *glnA* or *glnALG* gene deletions were generated in the *Y. pestis* EV strain by λRed-mediated mutagenesis [20] and confirmed by PCR (Table 2). The *Y. pestis* DNA fragment containing the respective deletion was then subcloned into the pCVD442 plasmid. The pCVD442-Δ*glnA*::*cat* or pCVD442-Δ*glnALG*::*cat* plasmid was transferred from an *E. coli* S17-1 *λpir* donor strain into a recipient wild-type *Y. pestis* strain 231 with the help of conjugation. The elimination of the suicide vector and isolation of *Y. pestis* clones of interest were achieved by culturing them in a medium containing sucrose and chloramphenicol, which acted as selective agents [24]. The resistance cassette was eliminated to produce FRT scar mutants by introducing pCP20 [23]. The presence of all *Y. pestis* virulence plasmids was confirmed via PCR amplification.

To construct the complemented strains, *Y. pestis* DNA was digested with BamHI and BglII (Thermo Fisher Scientific, Waltham, MA, USA). DNA fragments of an appropriate size were isolated from the gel, ligated with the pEYlpp vector treated with BamHI and FastAP alkaline phosphatase, and then transferred into DH5α cells. After DNA sequence verification, the recombinant plasmid pEYlpp-*glnALG* expressing *glnALG* was introduced into 231Δ*glnALG*, thus generating the complemented mutant strains C-231Δ*glnALG*.

### 2.4. Animal Challenges

*Y. pestis* 231 mutant strains were grown at 28 °C for 48 h on BHI plus 20 mM glutamine, which was diluted to an appropriate concentration in PBS. To demonstrate the loss of virulence and complementation of the mutant, groups of 6 outbred mice or 6 guinea pigs were challenged subcutaneously (s.c.) with serial tenfold dilutions of 231Δ*glnA* (10–10^4^ CFU for mice and 10–10^7^ CFU for guinea pigs), 231Δ*glnALG* (10^2^–10^5^ CFU for mice and 10^4^–10^7^ CFU for guinea pigs), C-231Δ*glnALG* (1–10^3^ CFU for mice and 10–10^4^ CFU for guinea pigs), and 231 (10–10^4^ CFU for mice and 10–10^4^ CFU for guinea pigs) at 0.2 mL aliquots. Survival was measured for 28 days post-inoculation. For vaccine studies, groups of 6 outbred mice or 6 guinea pigs were vaccinated via the s.c. route with serial tenfold dilutions of the Δ*glnALGY. pestis* mutant or only with a PBS buffer as a placebo. Four weeks after vaccination, the animals were injected s.c. with 200 LD_100_ (400 CFU for mice; 6 × 10^3^ CFU for guinea pigs) of the wild-type *Y. pestis* strain 231. All animals were observed over a 30-day period.

### 2.5. Antibody Titers

This study employed an indirect enzyme-linked immunosorbent assay (ELISA) to quantify the levels of immunoglobulin G (IgG) antibodies against *Y. pestis* antigens F1 [26,27] and LcrV [27,28] in mouse and guinea pig serum samples. Microtiter plates were coated with *Y. pestis* antigens at a concentration of 0.006 mg/mL and incubated overnight at 4 °C. Serum samples from mice and guinea pigs were then serially diluted and added to the wells. The endpoint titer, representing the highest dilution of serum yielding an absorbance reading 0.2 units above the background, was determined for each sample. Goat anti-mouse IgG conjugated with horseradish peroxidase (1:5000, Sigma, Saint-Louis, MO, USA) and goat anti-guinea pig IgG-HRP (1:5000, Sigma, Saint-Louis, MO, USA) were used as detection antibodies. After incubation with the detection antibodies, the reaction was developed using a TMB (3,3′,5,5′-Tetramethylbenzidine) substrate and stopped with sulfuric acid. Absorbance readings were measured at 450 nm. Background absorbance values were determined from serum samples obtained from animals injected with PBS alone.

### 2.6. Statistics

LD_50_ and the 95% confidence intervals of both mutated and virulent strains were determined for mice and guinea pigs using the Kärber method [29]. The graphs were generated using GraphPad Prism version 8.0.0 software for Windows (GraphPad Software, San Diego, CA, USA). Statistical analysis involved unpaired *t*-tests, ANOVA, and the Log-rank (Mantel–Cox) test. A significance level of *p* < 0.05 was established for all analyses.

## 3. Results

### 3.1. Genetic Organization of the glnALG Region

The genetic locus *glnALG* (Figure 1A) is located in *Y. pestis*, similar to that in *E. coli* and *Salmonella enterica*, between the *typA* and *hemN* genes. It was found to contain three open reading frames having the same transcriptional direction from *typA* to *hemN* and to be related to the gene clusters of *E. coli* and *S.* enterica (Figure 1B).

A high degree of similarity (91.5% identity) was shown between the predicted *glnA* gene sequence and the known *glnA* gene from *E. coli* strain K-12. Based on this strong homology, we hypothesized that the predicted *glnA* gene encodes glutamine synthetase. This enzyme plays a crucial role in nitrogen metabolism by catalyzing the conversion of glutamate and ammonia into glutamine, thus regulating intracellular nitrogen levels [30]. Two proteins encoded by *glnL* and *glnG* share high-level identity (86.5% and 90.0%) with the known sensor histidine kinase GlnL that senses and then transduces the nitrogen starvation signal and the DNA-binding transcription factor GlnG of *E. coli* strain K-12, respectively, which are two-component regulatory systems.

### 3.2. Effect of glnA and glnALG Deletion on the Growth of Mutant Strains

In enterobacteria, glutamine synthetase is the sole enzyme that can synthesize glutamine [31]. The study examined the development patterns of both the wild-type and mutant strains, Δ*glnA*, Δ*glnALG*, and C-Δ*glnALG*, on BHI agar and in BHI broth with and without glutamine. The Δ*glnA* and Δ*glnALG* mutants grew comparable to the wild-type strain when glutamine was added to the BHI agar. However, in BHI broth with glutamine, the Δ*glnALG* mutant displays a slight but obvious growth defect. No growth was observed without glutamine (Figure 2 and Figure 3).

Glutamine synthetase is the exclusive enzyme responsible for glutamine production in enterobacteria [31]. This study investigated the growth patterns of both wild-type and mutant strains (Δ*glnA*, Δ*glnALG*, and C-Δ*glnALG*) when cultured on BHI agar and in BHI broth, with and without the addition of exogenous glutamine.

These results indicate that the limited availability of glutamine caused by the absence of a functional glutamine synthetase explains the auxotrophy for this amino acid in Δ*glnA* and Δ*glnALG Y. pestis* strains.

### 3.3. Loss of the Entire glnALG Operon, Not Just the Single glnA Gene, Reduces Mutant Virulence

Glutamine is one of the two central products in the nitrogen assimilation process, functioning as an amide donor in most biosynthetic reactions [32]. Thus, a defect in glutamine biosynthesis and the potentially associated disturbance in nitrogen metabolism may significantly alter pathogen–host relationships.

The virulence of the constructed *Y. pestis* 231Δ*glnA,* 231Δ*glnALG* mutants, complemented mutant C-231Δ*glnALG*, and the parent fully virulent *Y. pestis* strain 231 was determined after subcutaneous inoculation in mice and guinea pigs (Table 3).

A comparative assessment of virulence in outbred mice and guinea pigs did not reveal any significant differences in the LD_50_ values of the wild-type strain and the GlnA^–^ mutant (Table 3). The strain with the deleted *glnALG* genes was avirulent for mice at a dose of 10^5^, and for guinea pigs, it was avirulent at a dose of 10^7^ CFU (the greatest administered doses) when injected subcutaneously (Figure 4). All animals showed no signs of disease during the observation period (21 days). After infection with the wild-type strain 231 in doses of 10–10^3^ CFU, mice died by days 4–6, and guinea pigs infected with 10^2^–10^3^ CFU died by days 10–12 of observation (Figure 4). The introduction of the recombinant plasmid pEYlpp-*glnALG* restored the virulence of the mutant strain 231Δ*glnALG* in the subcutaneous infection of outbred mice and guinea pigs (Table 3, Figure 4).

### 3.4. Humoral Immune Responses

The serum antibody responses to the two main immunodominant *Y. pestis* protective antigens (F1 and LcrV) [28] were determined via ELISA. The level of IgG antibodies to F1 and LcrV antigens in the blood of mice and guinea pigs was assessed on the 28th day after subcutaneous immunization with the studied *Y. pestis* 231∆*glnALG* strain (Figure 5). Both anti-F1 (*p* < 0.0001) and anti-LcrV (*p* < 0.05) IgG titers in mice increased in a dose-dependent manner when the levels of IgG induced by doses of 10, 10^2^, 10^3^, 10^4^, and 10^5^ CFU were compared. The differences in the levels of antibody responses to *Y. pestis* LcrV in guinea pig sera were insignificant (*p* > 0.05). A dose-dependent increase in the levels of antibody titers in guinea pigs was observed to the F1 antigen (*p* < 0.005).

### 3.5. Strain with Deletion of the Entire glnALG Operon Provides Protective Immunity Against Fatal Plague

Subcutaneous administration of 10^5^ CFUs of the 231Δ*glnALG* mutant strain did not cause death of mice for as long as 28 days post-inoculation, and all of the mice that survived such a challenge were fully protected against a subcutaneous inoculation of 200 LD_100_ of the wild-type strain (Figure 6). This indicated that the 231Δ*glnALG* mutant strain behaved like a live vaccine strain. When we inoculated lower doses (10^2^–10^4^) of the avirulent strain 231Δ*glnALG*, the survival of vaccinated animals after subcutaneous administration of the wild-type strain 231 decreased to 80–40%, demonstrating a dose-dependent protective effect. All PBS-treated mice in the control group died by day 4 post-infection. As in the case of mice, all guinea pigs treated with the 231Δ*glnALG* strain at doses of 10^4^–10^7^ survived subsequent infection with the virulent *Y. pestis* strain 231, which was not the case for guinea pigs treated with PBS (Figure 6).

## 4. Discussion

The main processes of life of any organism are growth and reproduction, which require energy sources and mineral substances. Pathogenic microbes use the host’s nutrients for these purposes, and the host resists infection by limiting the pathogen’s access to nutrients [33,34]. In turn, pathogens have developed strategies that combine efficient systems for absorbing nutrients synthesized in the host’s body with the ability to synthesize energy sources and plastic substances that are absent from their host. This area of host–pathogen interaction has been called nutritional virulence and has been the focus of increasing attention in the development of candidate vaccine strains [33].

Bacterial auxotrophs lack the metabolic capability to produce essential organic compounds. Consequently, they rely on their host organisms for these nutrients. Glutamine synthetase plays a crucial role in nitrogen assimilation by catalyzing the conversion of glutamate and ammonia into glutamine. This process is energy-dependent, requiring ATP as the fuel [35]. Glutamine plays a crucial role in the metabolic processing of nitrogen within bacteria, acting as a marker for intracellular nitrogen availability. When external nitrogen sources become scarce, the concentration of glutamine within bacterial cells diminishes, whereas the level of glutamate remains relatively [2]. Δ*glnA* mutant of *S*. Typhimurium mimics intracellular nitrogen limitation even when nitrogen is available in excess from the outside. Reduction in the glutamine pool under nitrogen limitation conditions is responsible for the slow growth of *S. enterica* [2]. The data above are consistent with our observations that *Y. pestis* strains lacking the *glnA* or *glnALG* genes do not grow on solid or liquid nutrient media, not even in those of a rich composition (BHI agar and broth) without the addition of glutamine. The addition of glutamine to the culture medium caused the ability of Δ*glnA* or Δ*glnALG* mutants to grow in vitro. Based on the growth data on nutrient media, it can be assumed that the ABC glutamine transporter encoded by the *glnHPQ* gene group functions in *glnA* or *glnALG* knockout mutants at a level necessary to ensure their nutritional needs at all stages of the plague pathogen life cycle in the host organism.

To definitively prove the role of a specific factor in causing the plague, we would need to show that removing this factor from a wild-type virulent strain of the bacteria reduces its ability to cause the disease. Furthermore, reintroducing the removed factor should then restore the bacterial virulence to its original level. We constructed a complementation plasmid pEYlpp-*glnALG*, which confirmed that the ablation of glutamine synthesis, along with the removal of the two-component regulatory system GlnL-GlnG, ensures significant attenuation of the mutant [12]. To demonstrate that a specific factor has a minimal impact on pathogenicity, it is sufficient to prove that removing this factor does not diminish the pathogen’s virulence [36]. In line with these considerations, we did not complement the knocked-out gene *glnA* because such a mutation did not decrease virulence. It was precisely this lack of effect on virulence that characterized the Δ*glnA* mutant.

The amide group present within the glutamine molecule serves as a nitrogen donor during the initial steps of synthesizing the fundamental units that compose essential cellular components such as proteins and nucleic acids. Therefore, it is expected that the observable characteristics of organisms with deletions in the *glnA* and *glnALG* genes will show notable distinctions when compared to organisms with the unaltered gene set. For example, according to Aurass et al. [14], the Δ*glnA* mutant of *S. enterica* serovar Typhimurium had a reduced ability to penetrate macrophages. It was shown that single Δ*glnA* mutants of *S. enterica* serovar Typhimurium did not have reduced virulence in intraperitoneally infected BALB/c mice [12]. While the deletion of the *glnA* gene alone did not significantly impact virulence, the simultaneous removal of both *glnA* and genes involved in glutamine transport (*glnH* and *glnQ*) or nitrogen regulation (*glnLG*) led to a notable decrease in the spread of these mutant strains within mice. Furthermore, these double knockouts also exhibited diminished survival rates within the J774 macrophages. This suggests that the degree of glutamine uptake by the host plays an important role in the development of the infectious process caused by *S. enterica* [12]. We also found that the single Δ*glnA* mutant of *Y. pestis* was not attenuated by the subcutaneous infection of mice and guinea pigs. However, the mutant with both the deletion of the glutamine synthetase gene (*glnA*) and the genes of the two-component nitrogen regulatory system (*glnLG*) dramatically reduced virulence during the subcutaneous infection of mice and guinea pigs. The obtained results confirm that the availability of glutamine for *Y. pestis* in the host organism depends on the two-component regulatory system *glnLG*, which, similarly for *S. enterica*, is apparently necessary for the transcription of glutamine transport genes [12]. Δ*glnA* bacteria also remain virulent because their genes that ensure the transport of glutamine into the cell and the genes that regulate this transport are not damaged. The glutamine transport system in Δ*glnAGL* mutants copes with ensuring their growth on nutrient media with a high glutamine content, but the damaged transport regulation system cannot cope with a full supply of glutamine in the host. 

Previous research indicated that the absence of the *glnGL* operon in the *Y. pestis* CO92 Orientalis strain did not diminish the bacterium’s ability to cause the disease [18]. This discrepancy in findings could be attributed to several factors, including differences in the bacterial strain (Orientalis versus Antiqua), the mouse model used (OF-1 versus a specific breeding line), and the method of infection (intradermal versus subcutaneous). These variables impact bacterial spread, immune responses, and ultimately, the severity of the disease. Further research involving controlled comparisons of these factors is necessary to elucidate the underlying reasons for the observed discrepancies.

Targeting genes essential for the production or transportation of crucial substances within microorganisms presents a viable strategy for developing vaccine strains effective against a wide range of infectious diseases. Knockout of genes responsible for the transport or synthesis of substances necessary for the vital activity of a microorganism is a promising approach in the creation of vaccine strains against many infectious diseases. We preliminarily assessed the immunogenicity of the constructed *Y. pestis* 231Δ*glnALG* strain in mice and guinea pigs. The Δ*glnALG* mutant did not cause death in mice when administered subcutaneously at 10–10^5^ CFU or in guinea pigs when administered subcutaneously at 10^4^–10^7^ CFU, and at doses of 10^5^ CFU for mice and 10^4^–10^7^ CFU for guinea pigs, it provided 100% protection of animals when subsequently administered subcutaneously with 200 LD_100_ of the virulent *Y. pestis* 231 strain. The immunogenicity of the 231Δ*glnALG* strain was more pronounced in the guinea pig model. Thus, the attenuated *Y. pestis* strain 231Δ*glnALG* can be considered a promising vaccine strain candidate. To make a final conclusion about the safety of the strain 231Δ*glnALG*, it is necessary to infect laboratory animals with large doses, 10^7^ CFU for mice and 2 × 10^9^ CFU and 1,5 × 10^10^ CFU for guinea pigs [37].

This type of attenuation of *Y. pestis*, associated with interference in the metabolism of the microorganism, may provide additional advantages in the construction of a vaccine strain, since the expression of the plague microbe genes associated with pathogenicity is not damaged and the complete antigen composition is available for recognition by the immune system and the formation of an immune response in the host.

## 5. Conclusions

In summary, the operon *glnALG* is functionally important in *Y. pestis*, and thus, its viability as a drug target should be explored.

In addition, a direction for further research may be to study the influence of mutations in genes whose products are responsible for the synthesis and high-affinity transport of nutrients on the pathogenicity of both intracellular and extracellular microorganisms. This requires constructing paired mutations in genes responsible for nutrient biosynthesis and transport or, alternatively, deleting genes of transport systems in auxotrophic pathogens that rely on the uptake of nutrients from the host organism.

Elucidation of host–pathogen interactions at the level of their metabolism should ultimately lead to a deeper understanding of the molecular mechanisms of the pathogenesis of bacterial infections and will allow the selection of optimal molecular targets for vaccine prophylaxis, as well as new pathogen-specific antimicrobial therapy strategies.

## Figures and Tables

**Figure 1 vaccines-13-00353-f001:**
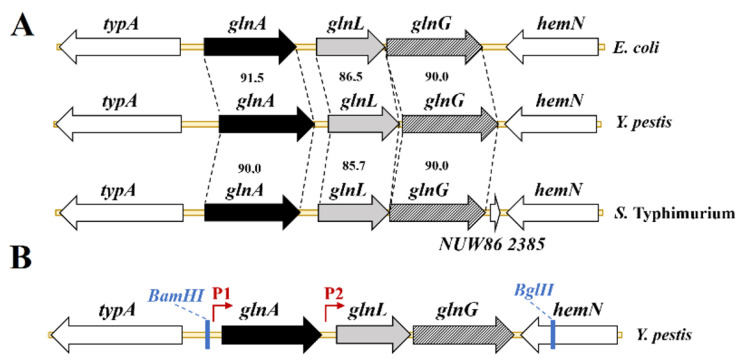
Schematic representation of the *Y. pestis glnALG* region. Comparison of the *Y. pestis* CO92 (GenBank accession number NZ_CP009973) genetic locus *glnALG* with those of *E. coli* strain K-12 substr. MC4100 (GenBank accession number HG738867) and *S. enterica* subsp. *enterica* serovar Typhimurium strain ATCC 14028 (GenBank accession number CP102669). The amino acid identities are shown between genes of a conserved gene order (**A**). Schematic representation of the *Y. pestis glnA* (P1) and *glnLG* (P2) promoters (**B**).

**Figure 2 vaccines-13-00353-f002:**
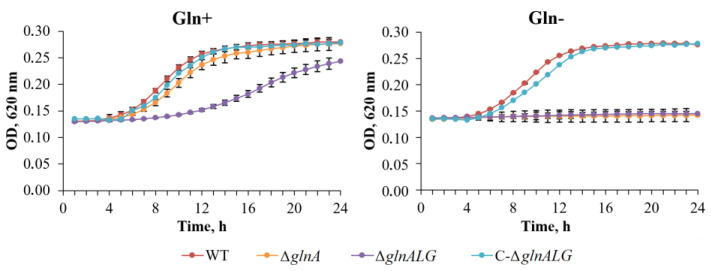
Glutamine requirement of the wild-type, Δ*glnA*, Δ*glnALG*, and C-Δ*glnALG Y. pestis* strains in broth culture. Cultures were inoculated into BHI broth or BHI broth supplemented with 20 mM of L-glutamine (Gln+) or not (Gln–). Growth was monitored by assaying the optical density (OD). Data are the means ± standard errors.

**Figure 3 vaccines-13-00353-f003:**
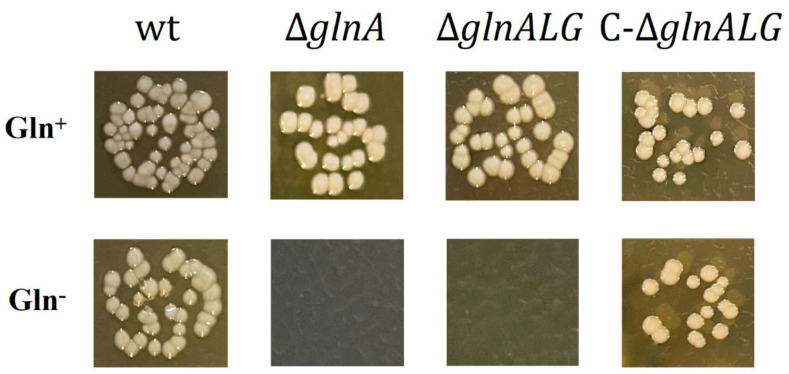
Growth of the wild-type, Δ*glnA*, Δ*glnALG*, and C-Δ*glnALG Y. pestis* strains on BHI agar supplemented with 20 mM of L-glutamine (Gln^+^) or not (Gln^−^).

**Figure 4 vaccines-13-00353-f004:**
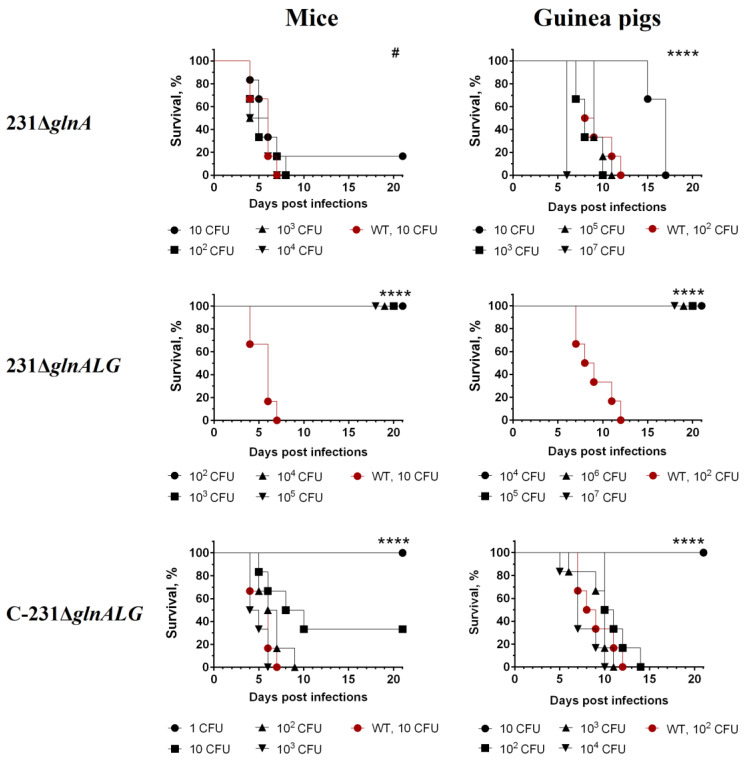
Survival of outbred mice (n = 6) and guinea pigs (n = 6) after subcutaneous infection with *Y. pestis* strains 231∆*glnA*, ∆*glnALG*, or C-∆*glnALG*. Log-rank (Mantel–Cox) test was used. #—*p* > 0.05; ****—*p* < 0.0001.

**Figure 5 vaccines-13-00353-f005:**
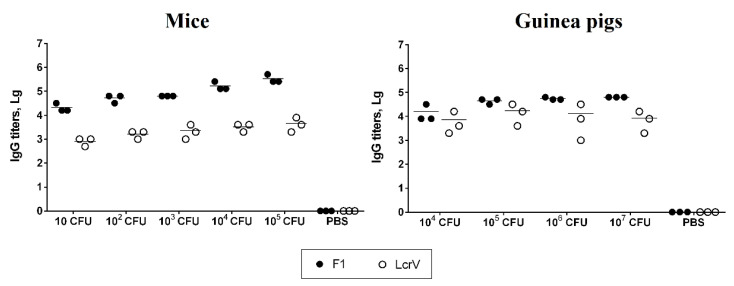
Anti-*Y. pestis* F1 and LcrV antibody titers in mouse and guinea pig sera collected 28 days after s.c. infection with 10, 10^2^, 10^3^, 10^4^, or 10^5^ CFU for mice or 10^4^, 10^5^, 10^6^, or 10^7^ CFU for guinea pigs of the *Y. pestis* 231 Δ*glnALG* mutant. Titers from three individual animals are shown; horizontal lines indicate the mean.

**Figure 6 vaccines-13-00353-f006:**
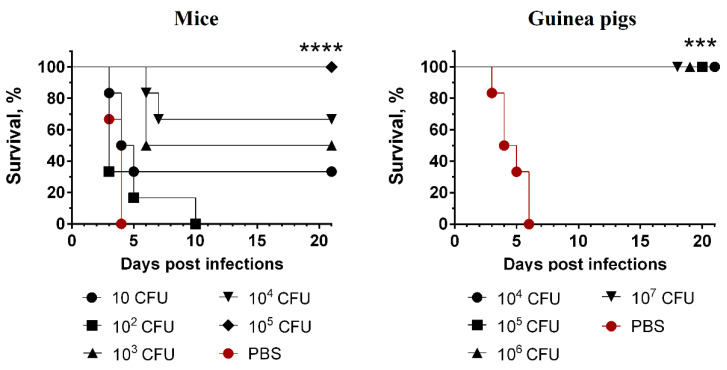
Survival of outbred mice and guinea pigs immunized subcutaneously with the *Y. pestis* 231Δ*glnALG* mutant strain after subcutaneous infection with 200 LD_100_ of the wild-type strain *Y. pestis* 231. Log-rank (Mantel–Cox) test was used. ***—*p* < 0.001; ****—*p* < 0.0001.

**Table 1 vaccines-13-00353-t001:** Bacterial strains and plasmids used in this study.

Strain, Plasmid	Relevant Attributes	Source
*Y. pestis*		
231	0.ANT3 phylogroup, wild-type strain, universally virulent (LD_50_ for mice ≤ 10 CFU, for guinea pigs ≤ 10 CFU); Pgm^+^, pMT1^+^, pPCP1^+^, pCD^+^	SCPM-O [21]
EV	1.ORI3 phylogroup, vaccine strain, Pgm^—^, pMT1^+^, pPCP1^+^, pCD^+^	SCPM-O
EVΔ*glnA*::*cat*	Δ*glnA* derivative of EV, Cm^r^	This study
EVΔ*glnALG*::*cat*	Δ*glnALG* derivative of EV, Cm^r^	This study
231Δ*glnA*	Δ*glnA* derivative of 231	This study
231Δ*glnALG*	Δ*glnALG* derivative of 231	This study
C-231Δ*glnALG*	Δ*glnALG* containing plasmid pEYlpp-*glnALG*	This study
*E. coli*		
S17-1 λ*pir*	*thi pro hsdR*^−^ *hsd*M^+^ *recA* RP4 2-Tc::Mu-Km::Tn*7*(Tp^R^Sm^R^Pm^S^)	SCPM-O
Plasmids		
pKD46	*bla* P_BAD_*gam bet exo*pSC101 *ori*TS	[22]
pKD3	*bla* FRT *cat* FRT PS1 PS2 *ori*R6K	[22]
pCP20	*bla cat c*I857 λP_R_*flp* pSC101 *ori*TS	[23]
pCVD442	*ori* R6K *mob* RP4 *bla sacB*	[24]
pEYR’	*ori* pA15 *cat* pR’	[25]
pEYlpp	*ori* pA15 *cat* p*lpp*	This study
pCVD442-*glnA::cat*	*ori* R6K *mob* RP4 *bla sacB cat glnA*	This study
pCVD442-*glnALG*::*cat*	*ori* R6K *mob* RP4 *bla sacB cat glnALG*	This study
pEYlpp-*glnALG*	*ori* pA15 *cat* Plpp *glnALG*	This study

**Table 2 vaccines-13-00353-t002:** Primers used in this study.

*glnA* Primers for Mutant Construction and Screening	
glnA1F	ATGCCTGAACACCATAAATGCAGTAACACACGGTAATCGTTCCACGACGACGACTATGGGAATTAGCCATGGTCC
glnA1R	GTGTTGGCTGCTTTCGCTCGCCACCTTCCTACACCTTGAAATCTATTAGGTAAACGTGTAGGCTGGAGCTGCTTC
glnA2F	CGGTCGCATCCAGGTTAACG
glnA2R	GCGTTACGGGTGATATTCAG
***glnALG* Primers for Mutant Construction and Screening**	
glnA1F	ATGCCTGAACACCATAAATGCAGTAACACACGGTAATCGTTCCACGACGACGACTATGGGAATTAGCCATGGTCC
glnLG3R	CTACTCCATCCCCAACTCTTTCAACTTCCGCGTTAATGTATTACGGCCCCAGCCCGTGTAGGCTGGAGCTGCTTC
glnA2R	GCGTTACGGGTGATATTCAG
glnLG2R	CTTGATTCTATTGCAACGGAAC
**Primers for pEYlpp Construction**	
Plpp-SphI	CGATGAGCATGCGATAACCAGAAGCAATAAAAAATC
PlppR-NdeI	CGATGTCATATGTAATACCCTCTAGTTTGAGTTAATC
**Screening for pCD1**	
yscFPlus	ACACCATATGAGTAACTTCTCTGGATTTACG
yscFMinus	ATTCTCGAGTGGGAACTTCTGTAGGATG
**Screening for pMT**	
caf1Plus	AGTTCCGTTATCGCCATTGC
caf1Minus	GGTTAGATACGGTTACGGTTAC
**Screening for pPst**	
PstF	CAATCATATGTCAGATACAATGGTAGTG
PstR	CTCCTCGAGTTTTAACAATCCACTATC

**Table 3 vaccines-13-00353-t003:** Virulence of *Y. pestis* strains in subcutaneously infected outbred mice and guinea pigs.

*Y. pestis* Strains	LD_50,_ CFU	
	Mice	GUINEA PIGS
231	1	15
231Δ*glnA*	5	3
231Δ*glnALG*	>10^5^	>10^7^
C-231Δ*gln*ALG	7	32

## Data Availability

All data will be provided upon reasonable request.

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
