# Peer review of "Protection Elicited by Glutamine Auxotroph of Yersinia pestis"

_vaccines, 2025, doi:10.3390/vaccines13040353_

Round 1
Reviewer 1 Report
Comments and Suggestions for Authors
The study conducted by Dentovskaya et al. presents findings on the safety and protective efficacy of two Yersinia pestis strains deficient in glutamine synthetase GlnA and the two-component regulatory system GlnLG, which controls ist expression, using animal models with mice and guinea pigs. The authors propose the attenuated Y. pestis strain 231 ΔglnALG as a promising vaccine candidate. The study design is appropriate; however, the manuscript lacks a brief introduction discussing other vaccine candidates and a more in-depth discussion of the results to strengthen its scientific impact. Nevertheless, several points require clarification and revision:
Specific Comments
- Introduction, lines 71-72. Please, provide a more detailed explanation of anti-nutritional-virulence therapy.
- What mouse strain was used for infection models? What was the rationale behind this choice?
- Material and Methods. Table 1 and line 122. The pEYlpp vector is missing from Table 1, and no reference to it is cited. Please clarify.
- Material and Methods. Section 2.5, line 138 and Results, line 218. For readers unfamiliar with Y. pestis, a brief mention of the F1 and LcrV antigens, used for antibody titers measurements, should be included. Additionally, clarify how these antigens were obtained or where they were purchased.
- Table 3. Please, provide a more detailed explanation of how LD50 values were determined using the Kärber method. What concentrations of wild type and mutant strains were tested? How many animals were used?
- Figure 4, line 216 and Figure 6, line 248. What statistical tests were used to analyze the survival curves?
- Figure 5. Antibody titers from only three animals is insufficient for obtaining significant results. Please justify or expand the dataset.
- Figure 4. Was there a significant difference between inoculation with 10 CFU of the mutant strain 231 ΔglnA compared to the wild type strain and the other mutant concentrations used for guinea pig inoculation?
- Results, section 3.5, lines 242-244. Why is the protective effect of the 231 ΔglnALG strain not dose-dependent in guinea pigs, unlike in mice? Please clarify.
- Discussion, lines 250-251. What types of “plastic substances” are required for essential life processes in organisms? Please specify.
- Discussion, lines 303-306. What is the basis to the statement: “The obtained results confirm that the availability of glutamine for Y. pestis in the host organism depends on the two-component regulatory system glnLG, which is apparently necessary for the transcription of glutamine transport genes”. Please, provide further explanation.
Minor Comments
- Table 1. In the “Relevant attributes” column for plasmids, please include the selection marker used for each plasmid.
- Results, lines 157-158. Please, clarify the following sentence: “…to contain 3 open reading frames having the same transcriptional direction from typA to hemN…” since the Figure 1A shows that the operon is transcribed in the opposite direction to typA and hemN.
- Legend of Figure 2. There are two inaccuracies: i) The statement “In all data points the error bars are smaller than the symbols” is incorrect, as it does not apply to all cases. ii) The statement “Growth of the wild-type strain was unaffected by the L-glutamine concentration” is misleading, as only one concentration (20 mM) was tested. Please revise accordingly.
- Line 312. Replace “stain” by “strain”.
- Introduction, lines 59-60. The sentence could be more concise since the two-component system was already defined earlier: “…also regulated by the two-component GlnLG system together with…”.
- Materials and Methods, lines 140-141. Since identical dilutions were used for both serum samples, the sentence could be reformulated as follows: “Both mouse and guinea pig serum samples were serially diluted from 1:250 to 1:512000”.
- Discussion, lines 252-253. Please, improve the term “food sources” for better clarity.
- Discussion, line 258. The term “obligate parasites” in reference to auxotrophic mutant strains is somewhat confusing in the context of bacterial strains. The authors may consider using a more precise term for auxothophs.
- Discussion, lines 275-278. Please, improve the clarity of the following sentence: “Direct proof of the importance of any determinant in the pathogenesis of plague would require demonstrating that a “wild-type” strain mutated in the gene encoding that virulence determinant would decrease its virulence, while complementation of the non-functioning gene would restore virulence to its former high level”.
- Discussion, lines 316-318. Please, enhance the clarity of the following sentence: “Knockout of genes responsible for the transport or synthesis of substances necessary for the vital activity of a microorganism is a promising approach in the creation of vaccine strains against many infectious diseases”.
Author Response
Dear reviewer, thank you very much for your thorough analysis and high evaluation of our manuscript. We have substantially revised it and made the following improvements and corrections in accordance to the requests:
Comment 1: The study design is appropriate; however, the manuscript lacks a brief introduction discussing other vaccine candidates and a more in-depth discussion of the results to strengthen its scientific impact. Nevertheless, several points require clarification and revision:
Response 1: Our review, Live Plague Vaccine Development: Past, Present, and Future (https://doi.org/10.3390/vaccines13010066), discussing other vaccine candidates and a more in-depth discussion of their merits and demerits, was published just in Vaccines on 13 January 2025. We have included a link to it in the introduction section.
Specific Comments
Comment 2: Introduction, lines 71-72. Please, provide a more detailed explanation of anti-nutritional-virulence therapy.
Response 2: A link to a publication devoted to nutritional virulence has been added to the text.
Comment 3: What mouse strain was used for infection models? What was the rationale behind this choice?
Response 3: Outbred means unpedigreed or underbred or not belonging to a certain strain. (i) First of all, outbred mice are cheaper. (ii) And, unlike purebred mice, they give a more heterogeneous response to vaccination and infection, which more accurately models the heterogeneous human population response.
Comment 4: Material and Methods. Table 1 and line 122. The pEYlpp vector is missing from Table 1, and no reference to it is cited. Please clarify.
Response 4: Added.
Comment 5: Material and Methods. Section 2.5, line 138 and Results, line 218. For readers unfamiliar with Y. pestis, a brief mention of the F1 and LcrV antigens, used for antibody titers measurements, should be included. Additionally, clarify how these antigens were obtained or where they were purchased.
Response 5: Added.
Comment 6: Table 3. Please, provide a more detailed explanation of how LD50 values were determined using the Kärber method. What concentrations of wild type and mutant strains were tested? How many animals were used?
Response 6: Added to Materials and Methods.
Comment 7: Figure 4, line 216 and Figure 6, line 248. What statistical tests were used to analyze the survival curves?
Response 7: Added.
Comment 8: Figure 5. Antibody titers from only three animals is insufficient for obtaining significant results. Please justify or expand the dataset.
Response 8: We acknowledge the reviewer's concern regarding the sample size. As stated in our manuscript, the statistical analyses revealed highly significant trends, particularly for anti-F1 IgG in mice (p < 0.0001) and a dose-dependent response (p < 0.005) in guinea pigs, despite the limited number of animals per group. These strong statistical differences indicate that the results are biologically meaningful. Furthermore, our study design adhered to the 3R principles, particularly by minimizing animal use while ensuring robust conclusions. Therefore, we respectfully suggest that our dataset is sufficient to support the conclusions drawn.
Comment 9: Figure 4. Was there a significant difference between inoculation with 10 CFU of the mutant strain 231 ΔglnA compared to the wild type strain and the other mutant concentrations used for guinea pig inoculation?
Response 9: There wasn’t significant difference between inoculation with 10 CFU of the mutant strain 231 ΔglnA compared to the wild type strain for guinea pigs.
Comment 10: Results, section 3.5, lines 242-244. Why is the protective effect of the 231 ΔglnALG strain not dose-dependent in guinea pigs, unlike in mice? Please clarify.
Response 10: One can ask us “Why is the protective effect of the 231 ΔglnALG strain not dose-dependent in guinea pigs, unlike in mice?” This is probably due to the fact that an order of magnitude smaller immunizing dose of the vaccine strain is required to form in guinea pigs immunity to plague of the same intensity as in mice. The ED50 value calculated for attenuated strains of the plague pathogen should not exceed 1×103 cfu for guinea pigs and 1×104 cfu for mice. The ED50 dose of a live vaccine that produces protection against death in 50% of the immunized guinea pigs was an order of magnitude less than the same indicator for mice [Anisimova, T.I.; Sayapina, L.V.; Sergeeva, G.M.; Isupov, I.V.; Beloborodov, R.A.; Samoilova, L.V.; Anisimov, A.P.; Ledvanov, M.Y.; Shvedun, G.P.; Zadumina, S.Y.; et al. [Main Requirements for Vaccine Strains of the Plague Pathogen: Methodological Guidelines MU 3.3.1.1113-02]; Federal Centre of State Epidemic Surveillance of Ministry of Health of Russian Federation: Moscow, Russia, 2002. [Google Scholar] [CrossRef]].
Comment 11: Discussion, lines 250-251. What types of “plastic substances” are required for essential life processes in organisms? Please specify.
Response 11: Replaced with “mineral substances”.
Comment 12: Discussion, lines 303-306. What is the basis to the statement: “The obtained results confirm that the availability of glutamine for Y. pestis in the host organism depends on the two-component regulatory system glnLG, which is apparently necessary for the transcription of glutamine transport genes”. Please, provide further explanation.
Response 12: A link to a publication devoted to explanation of this phenomenon has been added to the text.
Minor Comments
Comment 13: Table 1. In the “Relevant attributes” column for plasmids, please include the selection marker used for each plasmid.
Response 13: The selection marker for each plasmid included.
Comment 14: Results, lines 157-158. Please, clarify the following sentence: “…to contain 3 open reading frames having the same transcriptional direction from typA to hemN…” since the Figure 1A shows that the operon is transcribed in the opposite direction to typA and hemN.
Response 14: Replaced.
Comment 15: Legend of Figure 2. There are two inaccuracies: i) The statement “In all data points the error bars are smaller than the symbols” is incorrect, as it does not apply to all cases. ii) The statement “Growth of the wild-type strain was unaffected by the L-glutamine concentration” is misleading, as only one concentration (20 mM) was tested. Please revise accordingly.
Response 15: Corrected.
Comment 16: Line 312. Replace “stain” by “strain”.
Response 16: Replaced.
Comments on the Quality of English Language
Comment 17: Introduction, lines 59-60. The sentence could be more concise since the two-component system was already defined earlier: “…also regulated by the two-component GlnLG system together with…”.
Response 17: Done.
Comment 18: Materials and Methods, lines 140-141. Since identical dilutions were used for both serum samples, the sentence could be reformulated as follows: “Both mouse and guinea pig serum samples were serially diluted from 1:250 to 1:512000”.
Response 18: Done.
Comment 19: Discussion, lines 252-253. Please, improve the term “food sources” for better clarity.
Response 19: Done.
Comment 20: Discussion, line 258. The term “obligate parasites” in reference to auxotrophic mutant strains is somewhat confusing in the context of bacterial strains. The authors may consider using a more precise term for auxothophs.
Response 20: Done.
Comment 21: Discussion, lines 275-278. Please, improve the clarity of the following sentence: “Direct proof of the importance of any determinant in the pathogenesis of plague would require demonstrating that a “wild-type” strain mutated in the gene encoding that virulence determinant would decrease its virulence, while complementation of the non-functioning gene would restore virulence to its former high level”.
Response 21: Done.
Comment 22: Discussion, lines 316-318. Please, enhance the clarity of the following sentence: “Knockout of genes responsible for the transport or synthesis of substances necessary for the vital activity of a microorganism is a promising approach in the creation of vaccine strains against many infectious diseases”
Response 22: Done.
Reviewer 2 Report
Comments and Suggestions for Authors
The manuscript investigates the role of the glnALG operon in Yersinia pestis virulence and evaluates the ΔglnALG mutant as a live attenuated vaccine candidate. The study is well-structured, with clear objectives and methodologies. The findings contribute meaningfully to the understanding in bacterial pathogenesis and offer promising potentials for vaccine development. Following below are some suggestions to enhaunce readers' clarity.
Introduction: Write in more details the scope of the study.
Statistics. Add the statistical package used.
Discussion: The brief explanation for contrasting results with the CO92 strain (e.g., strain differences, infection routes) warrants further discussion. Elaborate on potential molecular or regulatory mechanisms that might underlie these differences.
Figures: Define all abbreviations in figure captions (e.g., Gln+ vs. Gln−)
Comments on the Quality of English LanguageMinor errors exist (e.g., "Say so, it has been reported..." → "For example, it has been reported..."). Performing language editing to correct grammatical errors is highly recommended.
Author Response
Dear reviewer, thank you very much for your thorough analysis and high evaluation of our manuscript. We have substantially revised it and made the following improvements and corrections in accordance to the requests:
Comments 1: Introduction: Write in more details the scope of the study.
Response 1: Added to the introduction “This study establishes that, Y. pestis defective for both glutamine synthetase GlnА and two-component sensor-transcriptional activator pair GlnL-GlnG completely lost virulence and provided potent protective immunity to mice and guinea pigs subsequently challenged with a wild-type Y. pestis strain, demonstrating glnALG operon potential use as a new molecular target for developing safe and efficient live plague vaccine.”
Comments 2: Statistics. Add the statistical package used.
Response 2: Added: "The graphs were prepared using GraphPad Prism version 8.0.0 software for Windows (GraphPad Software, San Diego, CA, USA)."
Comments 3: Discussion: The brief explanation for contrasting results with the CO92 strain (e.g., strain differences, infection routes) warrants further discussion. Elaborate on potential molecular or regulatory mechanisms that might underlie these differences.
Response 3: We think that several key differences between the studies—such as the bacterial strain, mouse strain, and inoculation route—complicate direct comparisons. These factors are known to influence infection dynamics and virulence outcomes, making it difficult to attribute the observed differences to a specific molecular or regulatory mechanism without further investigation. While such mechanisms may exist, addressing them would require additional targeted studies beyond the scope of this work. We have now clarified these points in the discussion to ensure transparency regarding the study's context and limitations.
Now it reads: “It has been reported that the loss of the glnGL operon in the Y. pestis CO92 Orientalis strain did not attenuate bacterial virulence [16]. This different result may be explained by variations in bacterial strain background (Orientalis vs. Antiqua), the mouse strain (OF-1 vs. our breeding line), and the route of infection (intradermal vs. subcutaneous) used in the two studies. These factors are known to influence bacterial dissemination, immune responses, and overall virulence outcomes. Further investigation, including controlled comparisons of these variables, will be needed to clarify the underlying reasons for the observed differences.”
Comments 4: Figures: Define all abbreviations in figure captions (e.g., Gln+ vs. Gln−)
Response 4: Done.
Comments 5: Minor errors exist (e.g., "Say so, it has been reported..." → "For example, it has been reported..."). Performing language editing to correct grammatical errors is highly recommended.
Response 5: Corrected.
Round 2
Reviewer 1 Report
Comments and Suggestions for Authors
The revised manuscript appears to have been prepared hastily, as it still contains several formatting errors (e.g., species and genes names not italicized, missing subscripts in chemical formulas, etc.) particularly in the newly added text.
Additionally, certain sections have been marked as revised despite no actual modifications being made. This has unnecessarily complicated the review process. The authors should clearly indicate the specific line numbers where changes have been implemented.
Regarding Response 3: while it is true that outbred mice better reflect the genetic and immunological diversity of natural populations, a larger sample size is required to obtain statistically significant and reproducible results.
Regarding Response 8: this reviewer insists that increasing the number of mice would be advisable, as this is a more appropriate approach for obtaining reliable and statistically significant antibody titer measurements. Given that the immunized mouse groups consisted of n = 6, it is unclear why the authors did not use serum samples from all animals in the group for the indirect ELISA assays (see Figure 5).
Regarding Response 10: the authors’ response does not sufficiently clarify the question raised.
Regarding Response 12 (Specific Comment 11 of the Reviewer): the reference added pertains to Salmonella Typhimurium, and this should be explicitly stated in the text. Furthermore, it remains unclear whether the results presented in this manuscript support the statement made.
Regarding Response 14 and Figure 1 (Minor comment 2 of the Reviewer): the three genes (glnA, glnL and glnG) share the same transcriptional direction, but are located on the complementary strand to hemN and typA, as indicated by the arrowhead in the figure. Is this correct?
Finally, no modifications appear to have been made in response to Comment 6 on the Quality of English Language (Response 22 of the authors).
Comments on the Quality of English Language_
Author Response
Comments 1: The revised manuscript appears to have been prepared hastily, as it still contains several formatting errors (e.g., species and genes names not italicized, missing subscripts in chemical formulas, etc.) particularly in the newly added text.
Response 1: We apologize to our reviewers for our inattention. We have made edits in accordance with the comments. Italics, superscripts and subscripts are highlighted in red letters. Species and gene names are now formatted in italics, and subscripts in the chemical formula are highlighted in red letters.
Comments 2: Additionally, certain sections have been marked as revised despite no actual modifications being made. This has unnecessarily complicated the review process. The authors should clearly indicate the specific line numbers where changes have been implemented.
Response 2:The line numbers where changes were implemented are indicated.
Comments 3: Regarding Response 3: while it is true that outbred mice better reflect the genetic and immunological diversity of natural populations, a larger sample size is required to obtain statistically significant and reproducible results.
Response 3: We share with our readers the preliminary results of our research. In the future, we plan full-scale preclinical studies according to [Anisimova, T.I.; Sayapina, L.V.; Sergeeva, G.M.; Isupov, I.V.; Beloborodov, R.A.; Samoilova, L.V.; Anisimov, A.P.; Ledvanov, M.Y.; Shvedun, G.P.; Zadumina, S.Y.; et al. [Main Requirements for Vaccine Strains of the Plague Pathogen: Methodological Guidelines MU 3.3.1.1113-02]; Federal Centre of State Epidemic Surveillance of Ministry of Health of Russian Federation: Moscow, Russia, 2002].
Comments 4: Regarding Response 8: this reviewer insists that increasing the number of mice would be advisable, as this is a more appropriate approach for obtaining reliable and statistically significant antibody titer measurements. Given that the immunized mouse groups consisted of n = 6, it is unclear why the authors did not use serum samples from all animals in the group for the indirect ELISA assays (see Figure 5).
Response 4: The main objective of our study was to find a new molecular target for attenuation of Y. pestis with the aim of generating an avirulent strain that protects from death animals subsequently infected with a virulent strain. The optimal way to control the safety and immunogenicity of the generated strain is to immunize laboratory animals with subsequent infection with a virulent strain. Obviously, assessing the effectiveness of plague vaccines in humans with their infection with virulent Y. pestis strains is not acceptable at all. The intensity of anti-plague immunity in humans is assessed based on immune response correlates such as an increase in antibody titers and formation of cellular immunity. The use of these correlates allows us to compare the data obtained on laboratory animals with the results of human immunization. However, immune responses to different antigens in different species and even breeds of animals differ, which requires careful extrapolation of data obtained on animal models to humans. Retro-orbital blood sampling in mice is not the most humane procedure. Therefore, given the relative information content and preliminary status of our studies, we minimized the number of mice subjected to this painful procedure.
Comments 5: Regarding Response 10: the authors’ response does not sufficiently clarify the question raised.
Response 5: Let's try to answer more simply. Beginning from the first half of the 20th century, a series of publications showed that Y. pestis strains attenuated by mutations in different genes differ significantly in their ability to protect from death animals of different species infected with virulent strains [Vaccines 2025, 13(1), 66; https://doi.org/10.3390/vaccines13010066, Dentovskaya SV, Ivanov SA, Kopylov PKh, Shaikhutdinova RZ, Platonov ME, Kombarova TI, Gapel'chenkova TV, Balakhonov SV, Anisimov AP. Selective Protective Potency of Yersinia pestis ΔnlpD Mutants. Acta Naturae. 2015 Jan-Mar;7(1):102-8. PMID: 25927007; PMCID: PMC4410401]. In our case, a dose-dependent response can be obtained by reducing the immunizing dose for guinea pigs below 1000 CFU.
Comments 6: Regarding Response 12 (Specific Comment 11 of the Reviewer): the reference added pertains to Salmonella Typhimurium, and this should be explicitly stated in the text. Furthermore, it remains unclear whether the results presented in this manuscript support the statement made.
Response 6: Changed to: The obtained results confirm that the availability of glutamine for Y. pestis in the host organism depends on the two-component regulatory system glnLG, which, as for S. enterica, is apparently necessary for the transcription of glutamine transport genes.
Comments 7: Regarding Response 14 and Figure 1 (Minor comment 2 of the Reviewer): the three genes (glnA, glnL and glnG) share the same transcriptional direction, but are located on the complementary strand to hemN and typA, as indicated by the arrowhead in the figure. Is this correct?
Response 7: Yes, that's true.
Comments 8: Finally, no modifications appear to have been made in response to Comment 6 on the Quality of English Language (Response 22 of the authors).
Response 8: English language editing has been completed.